# Effect of Nitric Oxide Pathway Inhibition on the Evolution of Anaphylactic Shock in Animal Models: A Systematic Review

**DOI:** 10.3390/biology11060919

**Published:** 2022-06-16

**Authors:** Maryam Alfalasi, Sarah Alzaabi, Linda Östlundh, Rami H. Al-Rifai, Suhail Al-Salam, Paul Michel Mertes, Seth L. Alper, Elhadi H. Aburawi, Abdelouahab Bellou

**Affiliations:** 1College of Medicine and Health Sciences, UAE University, Al Ain, United Arab Emirates; maryam_alfalasi98@hotmail.com (M.A.); sarahsz_@outlook.com (S.A.); 2National Medical Library, College of Medicine and Health Sciences, UAE University, Al Ain, United Arab Emirates; lostlundh@uaeu.ac.ae; 3Institute of Public Health, College of Medicine and Health Sciences, UAE University, Al Ain, United Arab Emirates; rrifai@uaeu.ac.ae; 4Department of Pathology, College of Medicine and Health Sciences, UAE University, Al Ain, United Arab Emirates; suhaila@uaeu.ac.ae; 5Department of Anesthesia and Intensive Care, University Hospital of Strasbourg, 67091 Strasbourg, France; paul-michel.mertes@chru-strasbourg.fr; 6Faculty of Medicine, EA 3072, Federation of Translational Medicine, University of Strasbourg, 67091 Strasbourg, France; 7Division of Nephrology and Vascular Biology Research Center, Beth Israel Deaconess Medical Center, Boston, MA 02215, USA; salper@bidmc.harvard.edu; 8Department of Medicine, Harvard Medical School, Boston, MA 02215, USA; 9Department of Pediatrics, College of Medicine and Health Sciences, UAE University, Al Ain, United Arab Emirates; e.aburawi@uaeu.ac.ae; 10Institute of Sciences in Emergency Medicine, Academy of Medical Sciences of Guangdong, Guangzhou 510060, China; 11Department of Emergency Medicine, Academy of Medical Sciences of Guangdong, Guangzhou 510060, China; 12Department of Emergency Medicine, Wayne State University School of Medicine, Detroit, MI 48201, USA

**Keywords:** nitric oxide, nitric oxide synthase, guanylate cyclase, cyclic guanosine monophosphate, anaphylactic shock

## Abstract

**Simple Summary:**

Anaphylactic shock (AS) is the most serious consequence of anaphylaxis, with life-threatening sequelae including hypovolemia, shock, and arrhythmias. The literature lacks evidence for the effectiveness of interventions other than epinephrine in the acute phase of anaphylaxis. Our objective was to assess, through a systematic review, how inhibition of nitric oxide (NO) pathways affects blood pressure, and whether such blockade improves survival in AS animal models. AS was induced in all included studies after or before drug administration that targeted blockade of the NO pathway. In all animal species studied, the induction of AS caused a reduction in arterial blood pressure. However, the results show different responses to the inhibition of nitric oxide pathways. Overall, seven of fourteen studies using inhibition of nitric oxide pathways as pre-treatment before induction of AS showed improvement of survival and/or blood pressure. Four post-treatment studies from eight also showed positive outcomes. This review did not find strong evidence to propose modulation of blockade of the NO/cGMP pathway as a definitive treatment for AS in humans. Well-designed in vivo AS animal pharmacological models are needed to explore the other pathways involved, supporting the concept of pharmacological modulation.

**Abstract:**

Nitric oxide (NO) induces vasodilation in various types of shock. The effect of pharmacological modulation of the NO pathway in anaphylactic shock (AS) remains poorly understood. Our objective was to assess, through a systematic review, whether inhibition of NO pathways (INOP) was beneficial for the prevention and/or treatment of AS. A predesigned protocol for this systematic review was published in PROSPERO (CRD42019132273). A systematic literature search was conducted till March 2022 in the electronic databases PubMed, EMBASE, Scopus, Cochrane and Web of Science. Heterogeneity of the studies did not allow meta-analysis. Nine hundred ninety unique studies were identified. Of 135 studies screened in full text, 17 were included in the review. Among six inhibitors of NO pathways identified, four blocked NO synthase activity and two blocked guanylate cyclase downstream activity. Pre-treatment was used in nine studies and post-treatment in three studies. Five studies included both pre-treatment and post-treatment models. Overall, seven pre-treatment studies from fourteen showed improvement of survival and/or arterial blood pressure. Four post-treatment studies from eight showed positive outcomes. Overall, there was no strong evidence to conclude that isolated blockade of the NO/cGMP pathway is sufficient to prevent or restore anaphylactic hypotension. Further studies are needed to analyze the effect of drug combinations in the treatment of AS.

## 1. Introduction

### Narrative Review

Anaphylaxis is a severe, rapid, systemic reaction to an allergen. Anaphylactic shock (AS) is the most serious consequence of anaphylaxis, with life-threatening effects on hemodynamics and cardiovascular function, including hypovolemia, shock and arrhythmias [1,2]. The manifestations are mainly caused by release of mediators of immune reactions involving IgE or non-IgE-mediated activation of mast cells and basophil activation [3,4].

The current lifetime prevalence of anaphylaxis is ~0.05–2% in the USA and ~3% in Europe [5]. The most frequently reported precipitants of AS are foods such as nuts, fish and shellfish, and drugs such as penicillin and its derivates, radiocontrast media, and anesthetics [6]. The European Academy of Allergy and Clinical Immunology (EAACI) defines food-induced allergy as a reproducible adverse reaction to food mediated by immunologic mechanisms involving IgE-mediated responses that often occur within hours of exposure [7].

It is recommended to administer intramuscular epinephrine as the first-line management of anaphylaxis to control hemodynamics, accompanied by supportive management [7]. The literature lacks evidence for the effectiveness of other interventions in the acute phase of anaphylaxis [6]. However, limited evidence supports coadministration of adjuvant therapies with intravenous epinephrine, including high flow oxygen, fluids, antihistamines, glucocorticoids, inhaled beta-2 agonists and inhaled epinephrine [6].

The pathophysiology of AS is complex and involves many organs. Histological changes in AS have been studied in animal models by Al-Salam et al. [8]. After AS, the lungs exhibit severe perivascular inflammation and edema, leading to a reduction in the alveolar space (see Figure 1) [8] and supporting involvement of NO in the pathophysiology of AS [8].

Additional research has implicated the possible involvement of IgG [9], platelet activating factor (PAF) [10,11], relaxin [12] and intestinal mast cell density [13], among other investigated factors. In this study, we focus on the role of inhibition of nitric oxide pathways (INOP) in the treatment of AS.

Studies of the role of NO in AS to date have proposed two major mechanisms. Firstly, mediators of anaphylaxis such as histamine, PAF, thromboxane A2 and leukotrienes are reported to stimulate NO release from the vascular endothelium [14], as summarized in Figure 2.

Activation of histamine receptors increases intracellular calcium, which binds calmodulin to activate endothelial NO synthase (eNOS) [2]. Activated eNOS transforms L-arginine to NO, which activates guanylyl cyclase to increase the concentration of cyclic guanosine monophosphate (cGMP). Increased [cGMP] activates cGMP-dependent protein kinases (PKGs), which leads to reduced cytosolic free calcium ([Ca^2+^]) and induces relaxation of vascular smooth muscle cells. This relaxation, in turn, leads to significant systemic vasodilation and, consequently, to hypotension [2]. In animal studies, mean arterial blood pressure (MAP) dropped by 65% within five minutes after induction of AS, leading to shock [8]. NO is produced by activation of the three isoforms of nitric oxide synthase (NOS), neuronal NOS (nNOS), endothelial NOS (eNOS) and induced NOS (iNOS), which differ in function and tissue distribution [14,15]. The constitutive nNOS and eNOS produce low amounts of rapidly metabolized NO with physiological roles in the regulation of arterial blood flow and blood pressure [16]. The inducible iNOS is synthesized de novo under the stimulus of inflammation [16].

Recently, we showed that iNOS and eNOS immunostaining increased after AS in pulmonary bronchial epithelial cells and in cardiac endothelial cells [8].

Cauwels et al. [10] used various strains of NOS-deficient animals to identify the roles of different NOS isoforms. Induction of AS in iNOS-deficient mice caused mortality comparable to control groups [10], whereas matched eNOS-deficient animals were significantly protected from shock [10]. eNOS seems to be a major enzyme in vasodilation, with a detrimental role in AS. It is proposed to target the PI3K/Akt/eNOS pathway to treat AS [10].

We have systematically examined relevant animal studies exploring how pharmacological blockade of NO pathways affects arterial blood pressure, and whether such blockade improves survival in AS animal models. Use of animal models to understand the role of NO in the occurrence of AS may allow exploration of new management options in humans.

## 2. Materials and Methods

### 2.1. Study Design

The review is registered online with the PROSPERO international prospective register of systematic reviews (CRD42019132273) and follows the Preferred Reporting Items for Systematic Reviews and Meta-Analyses (PRISMA) Statement [17]. The research question was created based on the Population, Intervention, Comparison, Outcome and Study design (PICOS) format:-Population: animal species with AS.-Intervention: blockade of NO production and guanylate cyclase activation.-Comparator: animal model with AS undergoing no treatment, epinephrine, or baseline measurements.-Outcomes: survival and normalization of blood pressure.-Study design: experimental.

### 2.2. Search Strategy

A comprehensive search of the literature including the biomedical databases PubMed (NLM), EMBASE (Elsevier), Scopus (Clarivate), Cochrane Library (Cochrane Collaboration) and Web of Science (Clarivate) was initially conducted on July 2019 by MA and SA in close collaboration with a medical librarian specialized in systematic reviews (LÖ). The complete search was updated in March 2022. Grey literature sources were not covered, as only peer-reviewed, published papers were considered for this review. PubMed and PubMed’s MeSH were used to systematically identify search terms and to develop a search strategy. The search term inclusion was reviewed by experts (AB and EHA) and the search string was peer reviewed by LÖ before it was adapted and applied to search all selected databases.

All search terms were searched in a combination of the search fields “article title”, “abstract” and “MeSH”/”thesaurus”. No filters or limitations to study design, publication dates or language were applied to the search, to ensure optimal information retrieval and to capture eventual pre-indexed materials. Hand screening of reference lists in the final papers was also conducted independently by MA and SA to ensure literature saturation. This yielded no additional references. Finally, the Predatory Reports from Cabell’s International [18] was consulted (MA and SA) to ensure the academic quality of the selected papers published in open access journals.

A search log with detailed search documentation and results for all databases is available in Appendix B. Appendix C and Appendix D are the PRISMA-S [19] and PRISMA 2020 checklists [17] for reporting, respectively.

### 2.3. Selection Process

All records identified in the literature search were uploaded to the systematic review software Covidence [20] for automatic de-duplication and blinded screening by two reviewers (SA and MA). A third reviewer (AB) resolved any conflicts reported by the software. The screening was done in two stages. Initial screening of titles and abstracts was conducted by two reviewers in accordance with the pre-specified inclusion and exclusion criteria.

The two reviewers then independently evaluated the full-text papers that matched the eligibility criteria and passed initial screening. An expert (AB) resolved any conflicts detected by the software. A PRISMA flow diagram with the details for the screening and selection process is available in Figure 3.

The pre-set eligibility criteria were used to determine whether the articles fitted the focus of the review, i.e., the association between NO and AS, and more specifically, the effect of INOP on survival and arterial blood pressure.

### 2.4. Eligibility Criteria

The inclusion criteria involved experimental studies (i.e., not reviews) published in full text (i.e., not abstracts or posters) with no restriction to language or year of publication. We included models of in vivo AS experiments with no restriction to animal species. This experimental model needed two groups to qualify: the experimental group with NO pathway inhibitors, and the control group, defined as any animals undergoing saline, no treatment, epinephrine or any other inactive substance.

Included outcomes are survival and/or changes in arterial blood pressure. Survival is measured from onset of AS until the end of the experiment. Studies that did not measure arterial blood pressure and/or survival were excluded.

Any other types of shock, in vitro, ex vivo and in silico models were excluded.

### 2.5. Data Extraction

An Excel sheet was designed based on data to be extracted, including details of the study design, animals, details including pre- or post-treatment, method, and duration of sensitization and AS, interventions, and our specified outcomes of interest. The primary outcome is the normalization of arterial blood pressure, measured in mmHg. The secondary outcome is animal survival, measured in minutes. The two reviewers independently performed data extraction from the published articles. Inconsistencies were resolved by discussion between the two reviewers.

In case of missing data, or data found only in graphs/diagrams but not in text, authors were contacted for the original data. Corresponding author emails were unavailable for some papers. Only one author of the four contacted regarding data for 11 articles responded to our request. This one author provided the requested data for three of the articles.

Data from unresponsive authors were extracted from published graphs using a PlotDigitzer software, version 2.6.8, SourceForge, San Diego, CA, USA [21]. If PlotDigitzer could not interpret the published graphs, that data was excluded from the analysis.

### 2.6. Risk of Bias

The SYstematic Review Center for Laboratory animal Experimentation (SYRCLE)’s risk of bias (RoB) tool for animal studies [22] was used to assess the methodological quality of the individual studies. An Excel sheet was created from the SYRCLE tool consisting of 10 “yes” or “no” questions (Appendix A). The two reviewers conducted the assessment separately, then resolved any disagreements. “Yes” indicated low RoB, “No” indicated high RoB, and “Unclear” indicated uncertain RoB. A “?” was used for items that were inapplicable to this study design. A summary of the risk of bias assessment is available in Figure 4.

The five studies (of 17) that included both pre-treatment and post-treatment models were assessed separately to ensure complete coverage in the assessment of bias.

For randomized allocation of animals (question 1), random number generator use was not mentioned in any of the studies. None of the studies mentioned the method of randomization therefore all were marked as of unclear risk.

Question 2 assessed confounder adjustment. All studies reported the same time between disease induction and intervention in experimental and control groups. In addition, the selected animals were littermates, with similar baseline characteristics such as species, sex, age and weight. This was determined to constitute a low risk of bias.

Nine studies mentioned randomized housing conditions (question 4). The rest were determined as unclear.

No studies described a method for randomized outcome assessment (question 6). Therefore, all were marked as unclear risk.

Regarding attrition bias (question 8), discrepancies in the numbers of animals were recognized in three studies. One study showed a missing animal in one of the experimental groups. In the other paper, one animal died during surgical preparation. The third study presented an extra animal in one experimental group in the graph as compared to the methods section. This study was marked as unclear.

There was no selective outcome reporting recognized for item 9.

Two additional potential sources of bias were assessed under item 10, contamination of drugs and conflicts of interest. No studies had added drugs that could be contaminants to the results. Fourteen studies reported no conflict of interest, and two papers were unclear.

The overall poor reporting in animal studies is a limitation for reliable assessment of the risk of bias. In several papers, important information regarding methodology was missing. For example, questions that refer to blinding/concealment (questions 3, 5 and 7), were inapplicable and not mentioned in any of the studies as they are not yet common practice in animal models [22]. This also meant that many assessments were marked as “Unclear RoB”. Therefore, it was difficult to reach a conclusion about the risk of bias across the studies.

### 2.7. Data Synthesis and Statistical Analysis

The purpose of our data extraction was to compile experimental data from different studies that address the role of NO pathways in AS animal models. However, data heterogeneity prevented a meta-analysis or even reliable statistical analysis. For example, studies using the same animal species used different medications. Even when stratified by medication tested, medication dosages and times of administration differed with respect to antigen challenge.

For all included studies, a qualitative assessment of results was performed.

## 3. Results

Overview

Nine hundred ninety unique studies were found in the literature search and screened for eligibility based on title/abstract screening against the pre-set inclusion and exclusion criteria. Of the 135 studies identified as eligible for full-text screening, 17 studies were finally selected for inclusion in the review [10,14,23,24,25,26,27,28,29,30,31,32,33,34,35,36,37]. A detailed PRISMA flow diagram regarding the screening and selection process is available in Figure 3.

### 3.1. Characteristics of Included Studies

First, we grouped the studies based on their status as pre-treatment or post-treatment studies. Five studies had experimental models for both [25,26,27,28,35]. We next organized the studies according to animal species. Two studies used dogs [29,30], five used rats [14,25,31,32,33], five used mice [10,23,24,34,37], three used pigs [26,28,35] and two studies used rabbits [27,36]. The characteristics are summarized in Table 1 and Table 2.

Shown in Figure 5 are the different medications used across the studies for inhibiting NO pathways. Note that some studies had experimental groups for more than one medication. Twelve studies used NG-nitro-L-arginine methyl ester (L-NAME) [10,14,23,24,25,29,30,31,32,34,36,37], seven studies used methylene blue (MB) [10,23,25,27,28,33,35], two studies used indigo carmine (IC) [25,26], one study used Aminoguanidine (AG) [32], one used 7-Nitroindazole (7–NI) [32] and two used 1H-[1,2,4] Oxadiazole [4,3-a] quinoxalin-1-one (ODQ) [10,23].

### 3.2. Descriptive Data Synthesis of the Effects of INOP on Arterial Blood Pressure and Survival

All included studies induced AS after or before medications that target blockade of NO production (Figure 5). In all animal species studied, the induction of AS caused a reduction in arterial blood pressure. However, the results showed different responses to the INOP.

To investigate the role of NO in animal models, several drugs with different mechanisms of action have been used across the studies. NG-nitro-L-arginine methyl ester (L-NAME) is a non-selective competitive NOS inhibitor that directly inhibits the biosynthesis of NO from L-arginine. Generally, direct NOS inhibitors are reversible once L-arginine is depleted [39]. Other drugs indirectly inhibit the NO pathway by targeting soluble guanylate cyclase (sGC). Methylene blue is a non-selective inhibitor of sGC in vascular smooth muscle [40] and, unlike L-NAME, offers the advantage of sparing some NOS-dependent physiologic effects [40]. 1H-[1,2,4] Oxadiazole [4,3-a] quinoxalin-1-one (ODQ) acts by inhibiting sGC, which disrupts NO-mediated signal transduction [10].

Indigo carmine (IC) inhibits endothelium-dependent vasodilation, affects peripheral alpha constrictors [25] and may inhibit sGC [41]. IC acts downstream of membrane receptors and involves cytosolic calcium [41]. Chang et al. [41] conclude that the site of action of IC is most likely NO synthase and/or to stabilize NO levels.

7-Nitroindazole (7-NI) is a relatively selective inhibitor of nNOS, but the mechanism underlying its effect in reversing AS is unknown. It has been assumed to counteract anaphylaxis-related sympathoinhibition, thus preserving vasoconstriction and attenuating antigen-induced hypotension [42].

Aminoguanidine hydrochloride (AG) is an inhibitor of iNOS, which is induced and regulated at the transcriptional level; therefore, AG is believed to modulate anaphylactic hypotension in the late phase [32].

### 3.3. Pre-Treatment

#### 3.3.1. L-NAME

L-NAME has been used in several experimental models to study its effect on AS. Studies on mice pre-treated before allergen challenge with L-NAME showed that the resulting NO blockade attenuated systemic hypotension and improved survival [10,23,24].

Some rat models exhibited gradual recovery from hypotension in both experimental and control groups, but the L-NAME prophylaxis groups had higher arterial blood pressure throughout the experiment [14,25].

Despite these positive results in some models, L-NAME has also been shown to be ineffective or to cause adverse effects. Studies in dogs [30] and rabbits [36] showed that pre-treatment with L-NAME caused no difference in hypotension in comparison to control groups. Similarly, no difference was observed for survival in rats [25]. L-NAME pre-treated rats had shorter survival times than control rats in studies by Bellou et al. [31] and Zhang et al., [32] and in rabbit studies by Mitsuhata et al. [36].

#### 3.3.2. Methylene Blue (MB)

Results of studies on rats by Albuquerque et al. [25] and Takano et al. [23] showed that pre-treatment with MB had no added protective effect in reversing hypotension. These findings reflect those in rabbits [23,27], pigs [28,35] and mice [10]. MB pre-treatment prolonged survival time in rats [25] and rabbits [27] but did not increase the survival rate.

#### 3.3.3. 1H-[1,2,4] Oxadiazole [4,3-a] quinoxalin-1-one (ODQ)

Cauwels et al. [10] showed that ODQ had no benefit in reversing shock in mice, in agreement with Takano et al. [23].

#### 3.3.4. Indigo Carmine (IC)

The use of IC before shock induction was shown to cause pronounced hypotension [26] and a worse survival rate compared with controls [25].

#### 3.3.5. Aminoguanidine (AG)

Zhang et al.’s [32] study on rats showed no improvement in hypotension or survival rates with the use of AG, an iNOS inhibitor, in comparison with the control group.

#### 3.3.6. 7-Nitroindazole (7-NI)

Zhang et al. [32] showed attenuation of hypotension with the use of the nNOS inhibitor, 7-NI, in rats, but survival rates were not improved in comparison with the control group.

### 3.4. Post-Treatment

#### 3.4.1. L-NAME

Use of L-NAME as a post-treatment after antigen challenge also showed contradictory results.

Reduced mortality was reported in mice [34]. Despite attenuation of hypotension in the dog model, treatment with L-NAME failed to improve survival [29]. L-NAME also worsened survival in rats [25].

#### 3.4.2. Methylene Blue (MB)

A study performed on rabbits showed a higher survival rate and restored arterial blood pressure compared with controls [27]. However, in other studies MB did not attenuate hypotension in rats [25,33] or in pigs [28,35]. In rats, one study showed that MB post-treatment reduced survival compared with control [25], while another study showed that a single bolus of MB significantly enhances survival time [33].

#### 3.4.3. Indigo Carmine (IC)

Albuquerque et al. studied IC on rats [25] and pigs [26]. They found that use of IC after shock induction caused exacerbation of hypotension throughout the experiment.

## 4. Discussion

NO has been shown to exert both protective and detrimental effects on the course and outcome of AS in animal models. Arterial blood pressure measurements in AS showed that the initial significant drop in blood pressure is clearly NOS/NO-independent [10]. Although the initial arterial blood pressure drop during AS was not different between L-NAME and control groups, the sympathoinhibition seen in the control group was counteracted in the L-NAME group [30]. After this initial arterial blood pressure drop, L-NAME pre-treated mice quickly recovered [10]. These data, in agreement with other studies [14], show that eNOS-dependent vasorelaxation plays a critical role in the pathophysiology of sustained hypotension and mortality in AS.

Nevertheless, NO has also been shown to have physiological benefit during AS, including bronchodilation, coronary artery vasodilation, decreased histamine release and anti-inflammatory properties [43]. However, while NOS inhibitors may improve arterial blood pressure, they also interfere with the cardioprotective effects of NOS and impair coronary circulation, causing a massive reduction in cardiac contractility and cardiac output [16,27].

Anaphylaxis-induced cardiac dysfunction and L-NAME-induced coronary vasoconstriction may synergize in causing left-sided heart failure with pulmonary congestion and edema, as shown in the postmortem examination by Zhang et al. [32]. Moreover, NO produced by the bronchial epithelium may play an important role in counteracting anaphylactic bronchoconstriction. NOS inhibitors may exacerbate bronchoconstriction in anaphylaxis and worsen the clinical condition [27].

The multiple mediators and metabolic pathways involved in anaphylaxis exhibit complex interactions. Pre-treatment with L-NAME caused a prostaglandin imbalance, with detrimental effects [31].

The use of different medications across the studies allowed exploration of the effect of different types of NOS. MB, an inhibitor of sGC activation, is also protective against shock, but not to the same degree as L-NAME [10]. This indicates an important, specific vasodilatory role for eNOS-derived NO, and suggests involvement of sGC-independent downstream mechanisms [10,33]. In support of this hypothesis, pre-treatment with the more specific sGC inhibitor, ODQ, showed even less protection against hypotension [10,23].

Cauwels et al. [10], Takano et al. [23] and Zhang et al. [32] used 7-NI and AG to study the respective effects of nNOS and iNOS inhibition, respectively. They showed that while 7-NI attenuated hypotension, AG did not, perhaps reflecting the requirement of hours rather than minutes for the transcriptional induction of iNOS [10].

In conclusion, NO and cGMP contribute to only one pathway involved in anaphylaxis, and isolated blockade of the NO/cGMP pathway is not sufficient to prevent or treat anaphylactic hypotension. Modulation of more than one AS pathway could be of interest in the treatment of AS. In the Zheng et al. study, combination of MB with EPI improves survival and arterial blood pressure, and prevents brain ischemia and neuronal apoptosis [33].

The literature describes case reports where methylene blue has been used in refractory shock states after the standard anaphylactic shock management failed [44,45,46,47,48,49,50]. In the majority of these cases, the hypotension resolved within 20 min [47]. Methylene blue’s availability and known doses make it easier to be used in clinical settings than other nitric oxide pathway inhibitors. The known side effects include nausea, vomiting and methemoglobinemia are not an issue considering the low dose used for anaphylaxis.

Early studies have shown that L-NAME causes a dose-dependent increase in systemic vascular resistance [51,52] and blood pressure in septic shock, and has a role in treatment of refractory cardiogenic shock [53]. No human studies were found that used L-NAME in anaphylactic shock.

New hypotheses have investigated potassium channel blockade [54,55] and inhibition of hydrogen sulfide pathways (our unpublished data).

Two papers have been published showing improvement of hypotension and survival in Wistar rats post-treatment using K^+^ channel blockers [54,55]. A possible mechanism is that K^+^ channels are involved in both endothelium-dependent and -independent vasodilation. Therefore, blocking these channels should help attenuate and promote recovery from shock. Another pathway was proposed by Tacquard et al., showing that blockade of platelet activating factor (PAF) receptor avoids decrease of left ventricular shortening function [11] and restores arterial blood pressure when combined with epinephrine.

Further experimental research should examine interactions between different signaling pathways to find a more effective treatment for AS.

## 5. Limitations of the Study

While animal models of AS provide a valuable tool to assess different parameters under controlled conditions, they do have limitations. Ethically, anesthesia must be used during experiments, but it is known to have effects on pathophysiological responses in AS.

These pathophysiological responses will also differ due to the genetic differences between animals and humans. Responses to medications also differ among animal species.

As discussed above, the lack of uniformity between study designs is also a limiting factor in interpreting the extracted results. The different medications, doses used, and the timing differences between pre-medication and antigen challenge to induce shock (see Table 1 and Table 2 for characteristics) could contribute to the variety of results seen. Pharmacokinetic and dose-response studies are missing.

Regardless of the shortcomings that limit comparability, preclinical animal studies of AS may provide important insights into possible treatments of AS in humans.

## 6. Conclusions

This review did not find strong evidence to propose modulation of blockade of the NO/cGMP pathway as a definitive treatment of AS in humans. Pre-treatment using inhibition of nitric oxide pathways showed improvement in BP and/or survival in seven out of fourteen experiments. When drugs were administered as post-treatment after the induction of AS, four out of eight experiments showed improvement of outcomes. 

Well-designed in vivo AS animal pharmacological models are needed. Other pathways are likely involved supporting the concept of pharmacological modulation using combinations of drugs.

## Figures and Tables

**Figure 1 biology-11-00919-f001:**
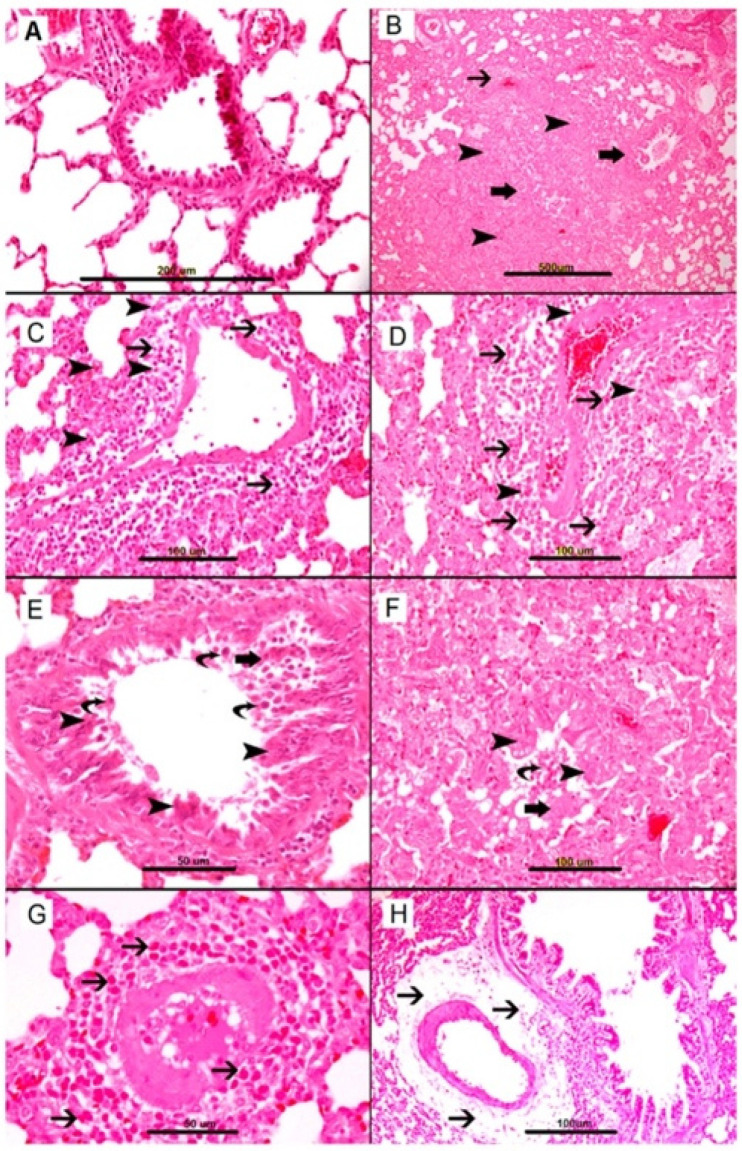
Representative sections of lung tissue. (**A**) Control group, showing lung tissue with patent alveolar spaces and bronchial passages and unremarkable blood vessels. (**B**–**H**) show anaphylactic changes in the lung. (**B**) Heavy mixed inflammatory cell infiltration of lung parenchyma with widening of interalveolar spaces (arrowheads), perivascular cellular infiltrates (thin arrows) and peribronchial inflammation (thick arrows). (**C**,**D**) Perivascular edema and heavy inflammatory cell infiltrate consisting predominantly of mast cells (thin arrows) and eosinophils (arrowheads). (**E**,**F**) Narrowing of the bronchial lumen with sloughing of respiratory epithelium (thick arrow), epithelial injury (arrowhead) and fallen dead cells in the lumen (curved arrow). (**G**) Heavy perivascular eosinophil infiltration (thin arrow). (**H**) Severe perivascular edema (thin arrow). This figure with explanatory text is obtained from an open access article distributed under the Creative Commons Attribution License which permits unrestricted use, distribution and reproduction in any medium, provided the original work is properly cited [8].

**Figure 2 biology-11-00919-f002:**
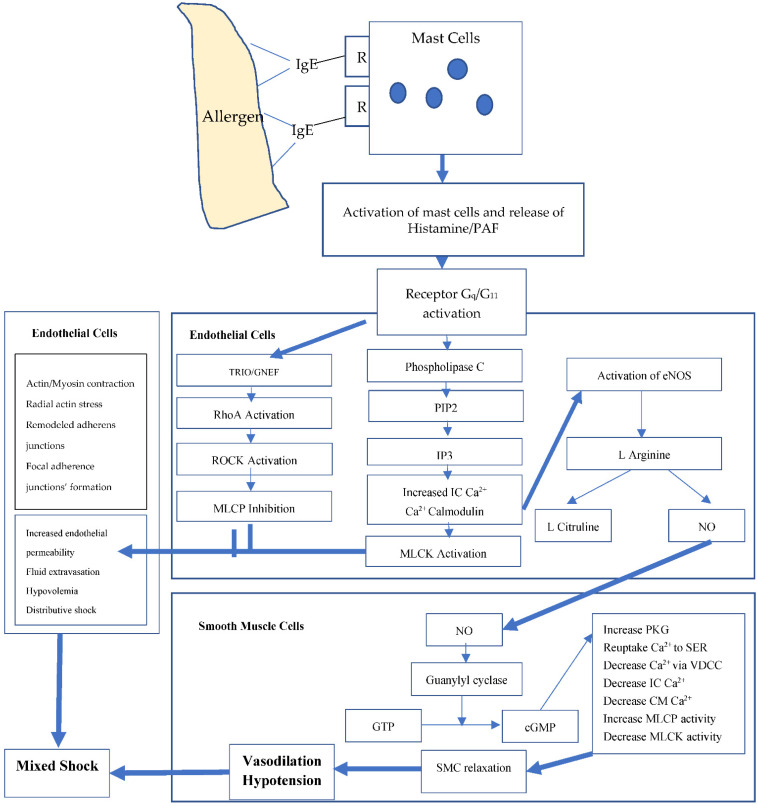
Binding of PAF and histamine to their respective receptors, which are G protein-linked, leads to activation of Gq/G11.This leads to the activation of the guanine nucleotide exchange factor, Trio, which in sequence activates small GTPases such as RhoA, which then activate the serine/threonine kinase, ROCK, which in turn phosphorylates myosin light chain phosphatase (MLCP), inhibiting its activity. Receptor binding activates phospholipase C which then catalyzes PIP2 (phosphatidyl inositol 4,5-bisphophate) hydrolysis to form DAG (diacyl glycerol) and IP3 (inositol triphosphate). Calcium-dependent activation of MLC kinase (MLCK) now occurs, resulting in increased actomyosin contractility and contributing to changing actin bundle orientation (induction of radial actin stress fibers), with the latter switching from being parallel to the junctions to perpendicular, thereby inducing junctional stress and disrupting integrity and vascular leakiness. Meanwhile, nitric oxide (NO), produced not by inducible nitric oxide synthase (iNOS), but by the constitutive endothelial form (eNOS) that is rapidly activated via the PI3K/Akt pathway. NO induces the formation of cGMP in smooth muscle cells from guanylyl cyclase (sGC), which then activates PKG, leading to a reuptake of calcium (Ca^2+^) from the cytosol by the sarcoplasmic reticulum (SER), as well as diminished calcium influx via voltage-dependent calcium channels (VDCC). This, combined with the opening of potassium channels and the exit of calcium from the cell leads to drop in intracellular calcium concentrations, inactivation of calmodulin and a resultant failure to activate MLCK. MLC phosphatase activity also increases correspondingly, leading to disruption of the actin–myosin cross-bridge and causing vasodilatation of blood vessels. Smooth muscle cells (SMC), immunoglobulin E (IgE), receptor (R). This figure was obtained from an open access article distributed under the Creative Commons Attribution License which permits unrestricted use, distribution and reproduction in any medium, provided the original work is properly cited. Authors slightly adapted the figure [4].

**Figure 3 biology-11-00919-f003:**
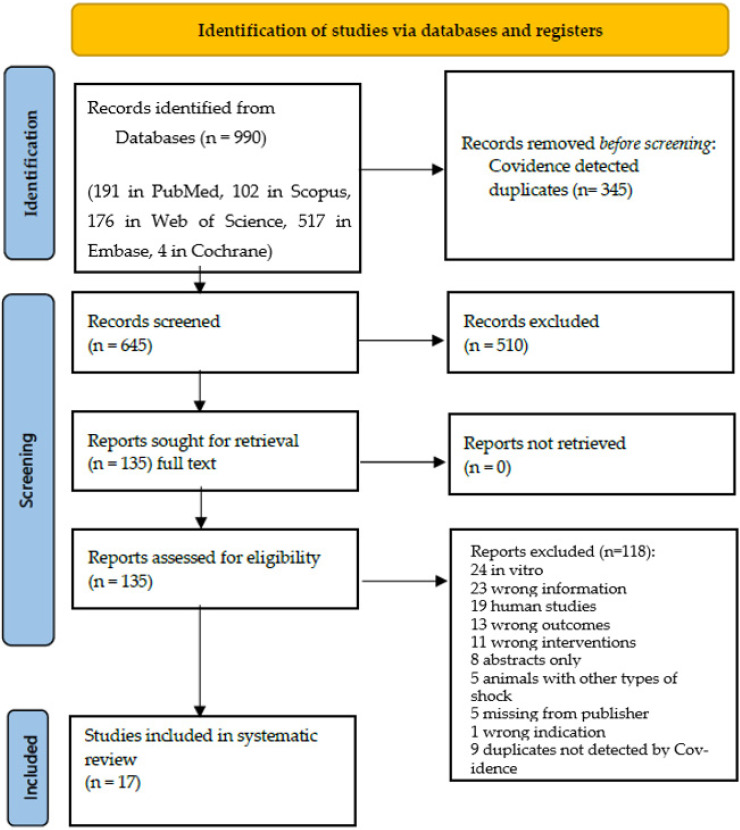
PRISMA 2020 flow diagram [17]. The number of studies identified from databases, screened for eligibility and the final included papers in the systematic review. This flow diagram is obtained from an open access article distributed under the Creative Commons Attribution License which permits unrestricted use, distribution and reproduction in any medium, provided the original work is properly cited [17].

**Figure 4 biology-11-00919-f004:**
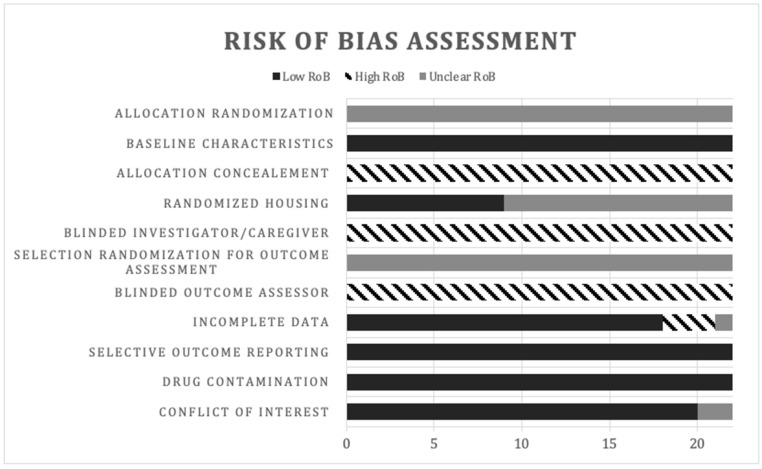
Evaluation of bias; risk of bias (RoB).

**Figure 5 biology-11-00919-f005:**
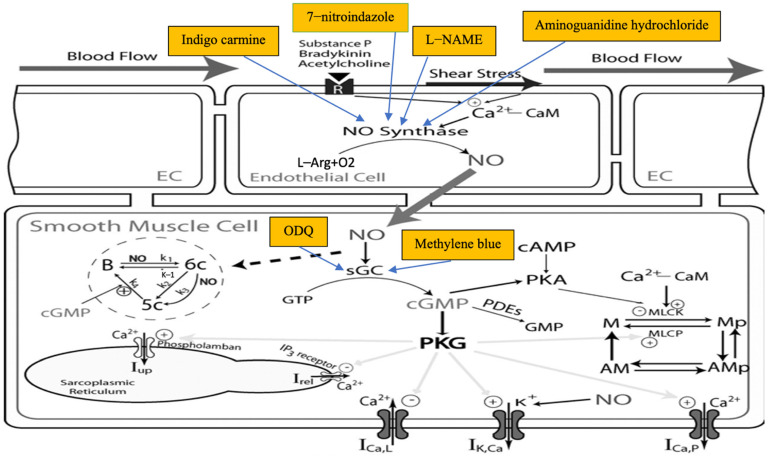
Molecular processes involved in the production of NO in endothelial cells and induction of vasodilation in smooth muscle cells.In response to environmental, neuronal, humoral or mechanical stimuli (e.g., ACh, bradykinin or shear stress), NO is synthesized in endothelial cells (EC) from l-arginine (l-Arg) by the activated form of endothelial NO synthase. NO diffuses to neighboring vascular smooth muscle cells (VSMC) where it activates soluble guanylate cyclase (sGC), which subsequently increases the intracellular cGMP production from GTP. B, basal form; 6c, 6-coordinate form; 5c, 5-coordinate form (fully activated). NO may directly (or via a pathway other than producing cGMP) regulate certain target proteins (e.g., Ca^2+^-activated K^+^ (KCa) channel). cGMP activates cGMP-dependent protein kinase (PKG), which regulates numerous target proteins, e.g., KCa channel current (IKCa), L-type Ca^2+^ channel current (ICaL), sarcolemma Ca^2+^-ATPase pump (ICaP), and myosin light chain phosphatase (MLCP), which leads to VSMC relaxation. cGMP is degraded into GMP by cyclic nucleotide phosphodiesterases (PDEs). Contractile kinetics shows that Ca^2+^ and cGMP co-mediated MLC phosphorylation and cross-bridge attachment. M, fraction of the free form of myosin light chain; Mp, fraction of phosphorylated myosin, AMp, fraction of myosin attached to actin filament; AM, fraction of attached myosin cross-bridges but dephosphorylated, namely, latch state. Total myosin is conserved, i.e., M + Mp + AMp + AM = 1. Note that VSMC is connected to EC via myoendothelial gap junctions. R, hormone receptors on EC membrane; CaM, calmodulin; MLCK, myosin light chain kinase; Iup, sarcoplasmic reticulum Ca^2+^ uptake current; Irel, sarcoplasmic reticulum Ca^2+^ release current; IP3, inositol 1,4,5-trisphosphate. Drugs blocking the production of NO (L-NAME, NG-nitro-L-arginine methyl ester; indigo carmine) and soluble guanylate cyclase, sGC (ODQ, 1H-[1,2,4] Oxadiazole [4,3-a] quinoxalin-1-one; methylene blue); cyclic adenosine monophosphate (cAMP). Aminoguanidine hydrochloride blocks inducible NOS. 7-Nitroindazole (7-NI) inhibits neuronal NOS. Copyright authorization for the figure and explanatory text re-use was requested through Copyright Clearance Center and granted by the American Physiological Society (APS). The paper is cited in reference [38]. We adapted the figure by adding the drugs that inhibit different NO pathways.

**Table 1 biology-11-00919-t001:** Pre-treatment study characteristics. Aminoguanidine hydrochloride (AG) anaphylactic shock (AS), bovine serum albumin (BSA), cyclic guanosine monophosphate (cGMP), dimethyl sulfoxide (DMSO), endothelial nitric oxide synthase (eNOS), heart rate (HR), histamine (H1), indigo carmine (IC), induced nitric oxide synthase (iNOS), intraperitoneal (IP), NG-nitro-L-arginine methyl ester (L-NAME), sodium chloride (NaCl), methylene blue (MB), nitric oxide (NO), nitric oxide synthase (NOS), neuronal nitric oxide synthase (nNOS), 1H-[1,2,4] Oxadiazole [4,3-a] quinoxalin-1-one (ODQ), ovalbumin (OVA), platelet activating factor (PAF), pulmonary arterial pressure (PAP), prostaglandin (PG), phosphoinositide 3-kinase (PI3K), systemic blood pressure (SBP), subcutaneous (SC), soluble guanylyl cyclase (sGC), 7–nitroindazole (7–NI).

Authors, Year	Title	Animal Species	AS Sensitization and Induction		Intervention and Dose		Pathophysiology Suspected
Osada et al., 1994 [37]	Participation of Nitric Oxide in mouse anaphylactic hypotension	ddY mice	Subcutaneous sensitization by 50 ug of hen egg-white lysozyme in Freund’s complete adjuvant on day 0. After 9 days, AS induced by 1 ug of intravenous (IV) lysozyme in saline		L-NAME 1 mg/kg 30 min before AS		Histamine released from sensitized mast cells stimulates vascular endothelial cells via H1 receptors. This leads to activation of NOS. The subsequent release of NO causes peripheral vasodilation through blood vessel smooth muscle stimulation, resulting in AS.
Mitsuhata et al., 1995 [36]	Nitric oxide synthase inhibition is detrimental to cardiac function and promotes bronchospasm in anaphylaxis in rabbits	Japanese white rabbits	Sensitized to horse serum with an initial 2 mL subcutaneous dose followed 2 days later by IV dose. After the second dose (14 days), AS induced by IV challenge with 2 mL horse serum over 10 s		L-NAME 30 mg/kg 15 min before AS		NOS inhibition may accentuate cardiac depression more than it increases venous return, therefore lowering the survival rate in L-NAME pretreated animals.
Shibamoto et al., 1996 [30]	Participation of nitric oxide in the sympathetic response to anaphylactic hypotension in anesthetized dogs	Mongrel dogs	Naturally sensitized to Ascaris antigen and shock induced by IV bolus 10 mg Ascaris suum diluted in 1 mL of saline		L-NAME 20 mg/kg bolus 15 min before anaphylactic shock and continuous infusion of 0.05 mg/kg per min (0.3 mg/min) over 75 min		NO is involved in the anaphylaxis-induced renal sympathoinhibitory response but not hypotension in anesthetized dogs.
Bellou et al., 2003 [31]	Constitutive nitric oxide synthase inhibition combined with histamine and serotonin receptor blockade improves initial ovalbumin-induced arterial hypotension but decreases the survival time in brown norway rats anaphylactic shock	Brown Norway Rats	SC 1 mg of OVA + 3.5 mg of aluminum hydroxide (Al OH) in 1 mL of 0.9% NaCl suspension given on day 0, 5 and 21. Shock induced by IV 1 mg OVA suspended in 1 mL of 0.9% saline		L-NAME, IV 100 mg/kg. 30 min before AS		Overall: imbalance between vasoconstrictor and vasodilator PG. NO synthase inhibition aggravates cardiac dysfunction and promotes bronchospasm. Inhibition of NOS3 by L-NAME could promote the activity of vasoconstrictor prostaglandins and/or leukotrienes, therefore decreasing HR by coronary vasoconstriction.
Buzato et al., 2005 [27]	The use of methylene blue in the treatment of anaphylactic shock induced by compound C48/80: experimental study in rabbits	New Zealand rabbits	No sensitization. AS induced by C48/80 intravenous bolus infusion (4/5 mg/kg)		MB 3 mg/kg intravenous bolus infusion 1–2 min before C48/80 infusion		The use of MB post-treatment reversed the AS hypotension but not when used as pre-treatment. Hypothesized pathophysiology involves the improvement of blood pressure by vasoconstriction. This proposes that MB has a role in increasing the smooth muscle cGMP, caused by NO released by histamine.
Cauwels et al., 2006 [10]	Anaphylactic shock depends on PI3K and eNOS derived NO	C57BL/6 mice	AS was induced by PAF. It was diluted in 200 μL endotoxin-free phosphate buffered saline (PBS) supplemented with 0.25% BSA and injected IV		L-NAME (Tempol was injected IP at 6 mg 1 h before PAF) 100 mg/kg IV	1 h before AS	The role of eNOS is important in regulating vascular function in shock. Downstream sGC is the main mediator for NO-induced vascular smooth muscle vasodilation.
						2 h before AS	
						4 h before AS	
					L-NAME (Tempol was injected IP at 6 mg 1 h before PAF) 100 mg/kg, IV, 2 h before AS		
					MB, in glucose solution suitable for IV injection at a dose of 15 mg/kg	1 h before AS	
						2 h before AS	
						4 h before AS	
						6 h before AS	
					ODQ was used i.p. in 50 μL DMSO at 20, 15, 10 or 5 mg/kg	0.5 h before AS	
						2 h before AS	
						4 h before AS	
			Single IP injection of 1 mg BSA mixed with 300 ng pertussis toxin. AS was induced 15 days after by:	BSA induced anaphylaxis with dose not mentioned	L-NAME, 200 mg/kg IV 2 h before AS		
				IV injection of 0.1 mg of BSA			
				IV injection of 2 mg of BSA			
Takano et al., 2007 [23]	NG-nitro-L arginine methyl-ester, but not methylene blue, attenuates anaphylactic hypotension in anesthesized mice	BALB/c mice	SC injection of an emulsion made by mixing aluminum potassium sulfate adjuvant (2 mg) with 0.01 mg ovalbumin, dissolved in saline (0.2 mL). The antigen emulsion was injected a second time, 7 days after the first antigen injection. AS was induced 1 week after the second injection. AS was induced by 0.01 mg of ovalbumin antigen (in 100 μL saline)		L-NAME, 1.0 mg/kg, 25 μL IV 10 min prior to AS		AS causes hepatic venoconstriction and portal hypertension, resulting in congestion of the upstream splanchnic organs. This decreases venous return and effective circulating blood volume exacerbates anaphylactic hypotension. L-NAME seems to increase systemic arterial blood pressure through sympathetic nerve activity stimulation of systemic arterioles but has no effect on hepatic circulation. Therefore, it was concluded that NO partially contributes to anaphylactic hypotension. The lack of improvement with MB or ODQ use suggests that there are sGC independent events downstream from NO production in AS that explain the beneficial effect of INOP.
					MB, 3.0 mg/kg, 25 μL IV 2 min prior to AS		
			Same as above but intraperitoneal for AS induction		ODQ, IP 10 mg/kg in 50 μL DMSO 1.5 h prior to AS		
			Naturally sensitized by C48/80, AS induced by C48/80 (4.0 mg/kg, 100 μL); IV		MB, 3.0 mg/kg, 25 μL IV 2 min before AS		
Zhang et al., 2009 [32]	7-Nitroindazole, but not L-NAME or aminoguanidine, attenuates anaphylactic hypotension in conscious rats	Sprague-Dawley rats	SC injection of an emulsion made by mixing equal volumes of complete Freund adjuvant (0.5 mL) with 1 mg ovalbumin dissolved in physiological saline (0.5 mL). Two weeks after, AS was induced by IV 0.6 mg of ovalbumin antigen in 300 μL saline		L-NAME, IV 10 mg/kg, 100 μL, 20 min before AS		7-NI (nNOS inhibitor) significantly attenuated the antigen-induced MAP decrease. Beneficial effect of 7-NI: nNOS inhibition might have counteracted the anaphylaxis-related sympathoinhibition, which preserved vasoconstriction of the resistance arteries and attenuated the antigen-induced systemic hypotension. L-NAME led to shorter survival time, most likely due to cardiac dysfunction and coronary vasoconstriction causing left heart failure and pulmonary congestion and edema. AG (iNOS inhibitor) did not affect the anaphylactic response.
					iNos inhibitor, IV Aminoguanidine hydrochloride (AG), 20 min before AS		
					nNos inhibitor, 7-Nitroindazole (7-NI), IP 50 mg/kg, 1 mL, 20 min before AS		
Menardi AC et al., 2011 [28]	Methylene blue administration in the compound 48/80-induced anaphylactic shock. Hemodynamic study in pigs	Dalland pigs	No sensitization. AS induced by bolus injection of C48/80 (4 mg/kg)		MB 2 mg/g bolus injection 3 min before AS		MB did not prevent or reverse the C48/80-induced anaphylactic shock; but the epidermal alterations did disappear after MB infusion. Pre-treatment had little to no effect on either.
Shinomiya et al., 2013 [24]	Nitric oxide and B2-adrenoceptor activation attenuate pulmonary vasoconstriction during anaphylactic hypotension in anesthetized BALB/c mice	BALB/c mice	Subcutaneous injection of an emulsion made by mixing aluminum potassium sulfate adjuvant 2 mg) with 0.01 mg ovalbumin dissolved in saline (0.2 mL). A second antigen injection was given 7 days after the first injection. The AS was induced one week after the second injection.		L-NAME 50 mg/kg; 50 μL 10 min before AS		Anaphylaxis causes pulmonary vasoconstriction, resulting in increased right heart afterload, and then a decrease in venous return, which finally contributes to anaphylactic hypotension. In this study, it was observed that L-NAME pre-treatment enhanced anaphylactic pulmonary vasoconstriction evidenced by the greater increases in systolic PAP.
Albuquerque AAS et al., 2016 [35]	Methylene blue to treat protamine-induced anaphylaxis reactions. An experimental study in pigs	Dalland pigs	No sensitization. AS induced by protamine IV infusion (dose not mentioned)		MB 3 mg/kg IV infusion (time not mentioned)		Protamine binds to an endothelial cell receptor that signals conversion of L-arginine to NO. NO activates sGC in the vascular smooth muscle to cause cGMP-mediated vasodilation. The resultant vasodilation decreases pulmonary vascular resistance and blood pressure. MB reversed the hypotension caused by protamine by acting on the NO/endothelium-dependent mechanism.
Mukai et al., 2018 [14]	Renal response to anaphylaxis in anesthetized rats and isolated perfused rat kidneys: roles of nitric oxide	Sprague Dawley rats	SC injection of an emulsion made by mixing equal volumes of complete Freund’s adjuvant (0.5 mL) and 0.5 mg ovalbumin. Two weeks after injection, shock induced by IV challenge with 0.6 mg of antigen		L-NAME, 10 mg/kg, 100 μL, IV. 10 min before AS		NO is produced in AS by different mechanisms that lead to hypotension and shock state. Proposed mechanisms of NO production are by the anaphylactic mediators inducing the vascular endothelium or by increased shear stress on the vascular endothelium. NO inhibitors reverse the AS by counteracting this hypotension.
Albuquerque et al., 2020 [25]	Effects of NO/cGMP inhibitors in a rat model of anaphylactoid shock	Male Wistar rats	Naturally sensitized by C48/80. AS was induced by C48/80 (3 mg/kg) IV bolus injection		L-NAME, 1 mg/kg IV 5 min before AS		The beneficial effect of L-NAME could be attributed to the blockage of eNOS. Removing NO production caused an SBP increase. MB is a non-selective GC inhibitor. When GC is inhibited, cGMP will not increase to cause vasodilation and hypotension. It is difficult to interpret the mechanism of IC’s effect on BP due to the ambiguous results.
					MB, 3 mg/kg 5 min before AS		
					IC, 3 mg/kg 5 min before AS		
Albuquerque et al., 2022 [26]	Indigo Carmine Hemodynamic Studies to Treat Vasoplegia Induced by Compound 48/80 in a Swine Model of Anaphylaxis	Male Daland Pigs	Naturally sensitized by C48/80.		IC 3 mg/kg 10 min before AS		IC inhibits endothelium-dependent relaxation specifically in relation to cGMP release. Additional effectiveness of IC was expected due to its alpha-adrenergic stimulation, which should counteract systemic hypotension. However, the vasoconstrictive effect was not apparent in this study.

**Table 2 biology-11-00919-t002:** Post-treatment study characteristics. Anaphylactic shock (AS), bovine serum albumin (BSA), cyclic guanosine monophosphate (cGMP), guanylyl cyclase (GC), indigo carmine (IC), intraperitoneal (IP), intravenous (IV), NG-nitro-L-arginine methyl ester (L-NAME), methylene blue (MB), nitric oxide (NO), nitric oxide synthase (NOS), peripheral vascular resistance (PVR), soluble guanylyl cyclase (sGC), Systolic blood pressure (SBP).

Authors, Year	Title	Animal Species	AS Sensitization and Induction	Intervention and Dose		Pathophysiology Suspected
Amir and English, 1991 [34]	An inhibitor of nitric oxide production, NG-nitro-L-arginine-methyl ester, improves survival in anaphylactic shock	Swiss Webster mice	IP with 2 mg bovine serum albumin (BSA) in 0.2 mL aluminum hydroxide gel. AS induced by IV 0.2 mL saline containing 100 ug BSA	L-NAME	30 mg/kg	The principal mediators of AS, histamine and bradykinin, stimulate NO release from vascular endothelial cells. NO relaxed vascular smooth muscle to cause venous dilation and systemic hypotension. Blocking NO production using L-NAME prevented vasorelaxation and improved the hypotension caused by AS.
					60 mg/kg	
			AS induced by C48/80		30 mg/kg	
					60 mg/kg	
Mitsuhata et al., 1995 [29]	An inhibitor of nitric oxide production, NG-nitro-L-arginine-methyl ester, attenuates hypotension but does not improve cardiac depression in anaphylaxis in dogs	Dog	Intradermal 0.1 mL of 1:100 dilution of an aqueous extract of Ascaris suum antigen with N2 concentration of 2.5 mg/mL. AS induced by 1 mL of A suum antigen into systemic circulation over 30 s	L-NAME 60 mg/kg (in 10 mL saline solution)		NO released by antigen challenge may be responsible (in part) for the hypotension due to vasodilation and fluid loss into the tissue space resulting from increased capillary permeability in anaphylaxis. NOS inhibitor did not improve cardiac function, which implies that production of NO in anaphylaxis may have a protective effect regarding cardiac function.
Buzato et al., 2005 [27]	The use of methylene blue in the treatment of anaphylactic shock induced by compound C48/80: experimental studies in rabbits	New Zealand Rabbits	AS induction by C48/80 IV bolus infusion (4.5 mg/kg)	MB 3 mg/kg venous bolus infusion		The use of MB post-treatment reversed the AS hypotension but not when used as pre-treatment. Hypothesized pathophysiology involves the improvement of blood pressure by vasoconstriction. This proposes that MB has a role in increasing the smooth muscle cGMP, caused by NO released by histamine.
Menardi AC et al., 2011 [28]	Methylene blue administration in the compound 48/80-induced anaphylactic shock: hemodynamic study in pigs	Dalland pigs	No sensitization. AS induced by bolus injection of C48/80 (4 mg/kg)	MB 2 mg/g bolus injection followed by continuous infusion of MB (2.66 mg/kg/h) delivered by syringe infusion pump		MB acts as an sGC inhibitor that abolishes the NO/cGMP-dependent smooth muscle vasodilatation.
Zheng et al., 2013 [33]	Methylene blue and epinephrine: a synergetic association for anaphylactic shock treatment	Brown-Norway rats	Sensitization by 1 mg grade VI chicken egg albumin (ovalbumin) and 4 mg aluminum hydroxide in adjuvant diluted in 1 mL 0.9% saline solution. Subcutaneous injection given on days 0, 4 and 14. AS induced on day 21 by IV injection of 1 mg ovalbumin.	A single bolus of 3 mg/kg MB		When MB was administered alone, there was disparity between the improved survival and the lack of tissue perfusion correction. This can be attributed to NO-independent pathway effects.
Albuquerque AAS et al., 2016 [35]	Methylene blue to treat protamine-induced anaphylaxis reactions. An experimental study in pigs	Dalland pigs	No sensitization. AS induced by protamine IV infusion (dose not mentioned)	MB 3 mg/kg IV infusion		Protamine binds to an endothelial cell receptor that signals conversion of L-arginine to NO. NO activates sGC in the vascular smooth muscle to cause cGMP-mediated vasodilation. The resultant vasodilation decreases pulmonary vascular resistance and blood pressure. MB reversed the hypotension caused by protamine, by acting on the NO/endothelium-dependent mechanism.
Albuquerque et al., 2020 [25]	Effects of NO/cGMP inhibitors in a rat model of anaphylactoid shock	Male Wistar rats	AS induction by C48/80 (3 mg/kg) IV bolus injection	L-NAME 1 mg/kg		The beneficial effect of L-NAME could be attributed to the blockage of eNOS. Removing NO production caused an SBP increase. MB is a non-selective GC inhibitor. When GC is inhibited, cGMP will not increase to cause vasodilation and hypotension. It is difficult to interpret the mechanism of IC’s effect on BP due to the ambiguous results
				MB 3 mg/kg		
				IC 3 mg/kg		
Albuquerque et al., 2022 [26]	Indigo carmine hemodynamic studies to treat vasoplegia induced by compound 48/80 in a swine model of anaphylaxis	Male Dalland Pigs	Naturally sensitized by C48/80.	IC 3 mg/kg 10 min after AS.		IC inhibits endothelium-dependent relaxation specifically in relation to cGMP release. Additional effectiveness of IC was expected due to its alpha-adrenergic stimulation, which should counteract systemic hypotension. However, the vasoconstrictive effect was not apparent in this study.

## Data Availability

Not applicable.

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
