# Peer review of "Effect of Nitric Oxide Pathway Inhibition on the Evolution of Anaphylactic Shock in Animal Models: A Systematic Review"

_biology, 2022, doi:10.3390/biology11060919_

Round 1
Reviewer 1 Report
With a real interest, I read the article biology-1739200. It is a very interesting article based on an elegant systematic search.
I have minor comments only:
1. The title: “Effect of Nitric Oxide pathway Inhibition on the Evolution of Anaphy-2 lactic Shock in Animal Models: A Systematic Narrative Review“. Well, usually either “systematic“ or “narrative“. Thus I would change to “Effect of Nitric Oxide pathway Inhibition on the Evolution of Anaphy-2 lactic Shock in Animal Models: A Narrative Review Based on a Systematic (Literature) Search“ or something like that.
2. Line 34: “All included studies induced AS after or before medications that target blockade of NO pathway.“. I would modify it, i.e. “AS was induced in all included studies after or before medications that target blockade of NO pathway were delivered.“ Or something like that.
3. Anyway, the article needs tob e adjusted tot he style oft he journal.
4. Figure 3: why are “191“ and “PubMed“ bolded?
5. Generally, the figures and their legends should be more uniform (and matching the style oft he journal).
6. Figure 5: please, make sure that all details are clearly visible. At the moment, some details are very, very small.
Reviewer 2 Report
The manuscript "Effect of Nitric Oxide pathway Inhibition on the Evolution of Anaphylactic Shock in Animal Models: A Systematic Narrative Review" by Alfalasi M et al. presents a narrative review based on 17 papers analyzing the possible role of inhibition of the NO pathway on anaphylactic shock. The review is interesting, but some issues need to be solved before publishing the paper.
All figures must be corrected from different points of view: title, legend, structure and even positioning in the text (correlated with the text where it was mentioned). Figure 1 (the second one - in fact, Figure 2) and Figure 5 must be redesigned to be easier to understand. Also, the Tables must be improved: Table 2 has no Title. The title of the papers and the journals may be omitted from Tables 1 and 2. All abbreviations used in the tables must be defined at the end as a legend.
The last paragraph from Results, "in summary," maybe better included in the conclusions.
The manuscript's structure must be improved: the Introduction is 3 pages long, more extended than the Results and far more extended than the Discussions (one page).
Round 2
Reviewer 2 Report
The authors made some changes based on the recommendations, but still, some changes are needed.
Regarding the legend of the figures. Fig 1 the following text should not be the first text, and there should be a title of the figure, then the explanation, and maybe the License for the image if needed.
"This figure with explanatory text is obtained from an open access article distributed under the Creative Commons Attribution License, which permits unrestricted use, distribution, and reproduction in any medium, provided the original work is properly cited."
The same comment is for Figure 2 and Figure 5.
The list of references needs some editing corrections. There is no need to add the database at the end of some references (PubMed, Embase, Scopus). No need for capital letters in reference no 44.
Author Response
We are grateful to Reviewer 2 for their time and effort:
We did all the edits requested in Figures 1, 2, and 5.
We corrected the references as pointed out.